# Polyimide Modified with Different Types and Contents of Polar/Nonpolar Groups: Synthesis, Structure, and Dielectric Properties

**DOI:** 10.3390/polym17060753

**Published:** 2025-03-13

**Authors:** Ting Li, Jie Liu, Shuhui Yu, Xiaojun Zhang, Zhiqiang Chen

**Affiliations:** 1Arrayed Materials (China) Co., Ltd., Shenzhen 518131, China; zircon.chen@arrayedmaterials.com; 2Shenzhen Institute of Advanced Technology, Chinese Academy of Sciences, Shenzhen 518055, China; j.liu2@siat.ac.cn (J.L.); shuhui.y1@siat.ac.cn (S.Y.)

**Keywords:** polyimide films, co-polyimide, energy storage, polarization mechanism, dielectric properties

## Abstract

Polyimide-based dielectric materials, as excellent high-temperature-resistant polymers, play a crucial role in advanced electronic devices and power systems. However, given the limitations of composite PI materials, it is a significant challenge to simultaneously improve the dielectric constant and breakdown strength of intrinsic polyimide dielectric materials to achieve high energy density. In this study, an indiscriminate copolymerization method was proposed, which utilizes two different diamine monomers with bulky side groups (-CF3) and high polarity (C-O-C), respectively, to copolymerize with the same dianhydride monomer and prepare a series of intrinsic PI films. Remarkably, PI films with a highly dipolar rigid backbone maintain excellent thermal and mechanical properties while enhancing dipole polarization. Meanwhile, a high breakdown strength of PI is shown, because the bulky side groups act as deep traps to capture and disperse charges during the charge transfer process. Under the optimal copolymer ratio, the dielectric constant and dielectric loss are 4.2 and 0.008, respectively. At room temperature, the highest breakdown strength reaches 493MV/m, and the energy storage density and charge–discharge efficiency are 5.07 J/cm^3^ and 82%, respectively. Finally, based on density functional theory calculations, the copolymerization tendencies of the three monomers are verified, and it is speculated that the copolymerization ratio of PI-60% is the most stable and exhibits the best overall performance, which perfectly aligns with the experimental results. These experimental results demonstrate the exciting potential of intrinsic polyimide in thin film capacitors.

## 1. Introduction

The ongoing trend toward miniaturization and weight reduction in power electronic systems has significantly increased the demand for capacitor polymer films that exhibit both high energy storage density and superior thermal stability [1,2,3,4,5]. Among various polymeric materials, polyimide (PI) has emerged as a prominent candidate for high-temperature energy storage dielectric applications due to its exceptional combination of mechanical strength, electrical insulation properties, and thermal stability [4,6]. Recent advancements in nanocomposite technology have demonstrated that the incorporation of nanoscale fillers into polymer matrices can substantially enhance energy storage performance [4,7,8]. However, precise control over the microstructural characteristics of fillers, including their dimensional parameters, morphological features, dispersion homogeneity, and void defect distribution, remains a significant technical challenge [9]. Furthermore, the inherent dielectric mismatch between the constituent phases induces electric field redistribution and facilitates charge injection at the electrode interface, which consequently deteriorates the breakdown strength of the composite system and ultimately compromises its energy storage capabilities [10,11,12]. These fundamental limitations pose substantial barriers to the industrial-scale production and practical implementation of nonlinear polymers incorporating multidimensional fillers [13,14,15,16,17,18]. Consequently, there exists a critical need to develop innovative, cost-effective, and scalable fabrication methodologies that can effectively mitigate both electrode charge injection and bulk carrier transport in dielectric polymers, particularly for demanding high-temperature applications [19].

Advancement in polymer-based energy storage materials has been significantly facilitated by the emergence of all-organic composite architectures [10,17]. One of the most common methods to improve the energy storage performance of PI is through the modification of its molecular structure. Among various modification strategies, molecular structure engineering has been widely recognized as one of the most effective approaches to enhance the energy storage characteristics of PI. This methodology primarily focuses on the systematic manipulation of polymer chain configuration, functional group substitution, and intermolecular interactions, which collectively influence the dielectric properties and energy storage performance of the resulting materials [20]. Zhu et al. [21] designed and synthesized a novel diamine monomer containing two -CN groups at the ortho position of the benzene ring, resulting in a large dipole (8.60 D) moment in the same direction. The prepared PI exhibited an improved dielectric constant (ξr) and maximum discharge energy density (Ue) of 4.80 and 1.023 J/cm^3^, respectively. Subsequently, Yuan et al. [17] reported the development of a fully organic composite composed of PEI (polyetherimide) and PCBM (derivative of fullerene). The composite material leverages the strong electrostatic attraction of PCBM to immobilize free electrons and hinder the injection and transport of charges within PEI. The composite material exhibits a high energy density of 3.0 J/cm^3^ and a high discharge efficiency of 90% at 200 °C. Wu et al. [22] synthesized a conjugated benzene amine trimer (ACAT) and attached it to the main chain of PI through in situ polymerization. This conjugated structure significantly improved the electronic polarization of the molecular chains and carrier mobility. In all, to enhance the dielectric and energy storage properties of polymer materials, modifications have been introduced to the molecular chains through the grafting of various polar groups or by copolymerizing with different monomers, notably polar dipoles possessing substantial dipole moments, such as -OH, -CN, -NH_2_, -SO_2_, and -NO_2_ groups. These groups are strategically grafted onto either the main chains or the side chains of the polymers with the aim of significantly improving the dielectric constant [23].

In addition to chemical modification, the physical blending approach for fabricating all-organic polyimide composites has demonstrated remarkable effectiveness in enhancing dielectric properties. The heterogeneous interface formed between the constituent polymers facilitates interfacial polarization through the suppression of premature polarization saturation. Furthermore, the interpenetrating network morphology significantly reduces charge carrier density within the polymer matrix, thereby minimizing void defect formation. This unique structural characteristic contributes to substantial improvement in dielectric breakdown strength (Eb), making it a promising approach for developing high-performance energy storage (Ue) materials. Ahmad et al. [24] filled poly (arylether urea) (PEEU) into a PI matrix, which showed a maximum Ue of 5.14 J/cm^3^ with Eb~495.65 MV/m at room temperature. Wu et al. [18] modified PP (polypropylene) and synthesized a copolymer with flexible hydroxyl (-OH) and polar functional groups (-NH_2_). When the hydroxyl content was 4.2 mol%, the dielectric constant of the modified PP increased to twice that of BOPP (biaxially oriented polypropylene), while maintaining a linear reversible polarization curve. At 600 MV/m, Ue was about 7 J/cm^3^. Xiao et al. [25] used random copolymerization to prepare PI separation membrane materials. The results showed that after copolymerization, a mesh-like structure formed between the polymer chains, and extended conjugated aromatic structures were generated between the units arranged within the polymer backbone, effectively increasing the internal cross-linking density of the polymer. The strategic design of molecular networks through precise engineering approaches has enabled the simultaneous optimization of mechanical strength and toughness, facilitating the development of advanced materials with tunable mechanical properties. Recent investigations employing ionic or crystalline domain cross-linking methodologies have successfully fabricated densely interconnected molecular architectures [26,27]. These sophisticated structures endow materials with precisely controllable mechanical characteristics, including an adjustable tensile strength and elastic modulus, while significantly expanding their potential applications across various engineering fields. The implementation of such cross-linking strategies has demonstrated remarkable potential in overcoming the traditional strength–toughness trade-off, thereby opening new avenues for the development of multifunctional materials with tailored mechanical performance.

To date, molecular modification has demonstrated tremendous potential for enhancing the dielectric and energy storage performance of intrinsic PI. Notably, in the design of high-performance dielectric energy storage materials, it is necessary to optimize the effectiveness of modification methods, geometric scales, matrix compatibility, and microstructural positions, among other multidimensional factors [5]. More significantly, computer-aided molecular design represents a milestone in industrial development. Despite the significant success of machine learning in predicting structure–property relationships in molecular and materials research, the establishment of polymer energy storage–dielectric property models is still in its early stages [28,29,30]. In fact, research on the breakdown of composite dielectrics often relies on various types of analysis variables (such as dielectric performance, energy gap, and structural morphology), and it is important to establish multi-variable coupling relationships to predict the energy storage–dielectric properties of materials.

In this study, we systematically investigated the copolymerization of two structurally distinct diamine monomers with a common dianhydride monomer through controlled random polymerization. The selected diamine monomers were specifically chosen to incorporate contrasting functional moieties: one containing a bulky trifluoromethyl (-CF_3_) group to introduce steric hindrance effects, and the other featuring a polar ether linkage (C-O-C) to enhance molecular interactions. The molar ratios of these diamine monomers were precisely adjusted according to a predetermined protocol while maintaining a stoichiometric balance between the total diamine and dianhydride monomers throughout the polymerization process. Remarkably, the introduction of a highly dipolar rigid backbone enabled the copolymerized PI to maintain excellent thermal and mechanical properties while enhancing dipole polarization. Thus, the dielectric constant was 4.2 with a loss of less than 0.008, at 1000 Hz. Bulky side groups were incorporated, acting as deep traps to capture and disperse charges during the charge transfer process, thereby improving the breakdown strength of the polymer. At room temperature, the energy storage density and charge–discharge efficiency were 5.07 J/cm^3^ and 82%, respectively. Importantly, based on DFT (density functional theory) calculations, a more significant structure with the best micro-electron structure and properties was shown, which related to energy levels and nearby functions. It was speculated that the copolymerization ratio of PI-60% was the most stable and exhibited the best overall performance, which perfectly matched the experimental results. Meanwhile, UV-vis-NIR was performed to verify the reaction tendencies. All in all, the experimental results provide new insights for the preparation of novel polymer materials with high energy storage density.

## 2. Experimental Section

### Synthesis and Preparation of PI Films

The synthesis routes of the PI films and the compounding ratios of the two diamine monomers under different copolymerization ratios are shown in Figure 1 and Appendix A, respectively. Under a nitrogen atmosphere, bis(4-(3-aminophenoxy)phenyl)methane (BABP), 2,2′-bis(trifluoromethyl)-4,4′-diaminobiphenyl (TFMB), dianhydride 3,3′,4,4′-benzophenone tetracarboxylic dianhydride (BPDA), and N, N-dimethylacetamide (DMAc) were added to a three-neck flask, and the mixture was stirred at a low speed in an ice bath. The reaction was then stirred for 22 h to obtain a polyamide acid (PAA), PAA samples with seven different copolymerization ratios were prepared (Appendix A), and the weight-averaged molecular weights (Mw), number-averaged molecular weights (Mn), and polydispersity index (PDI) of the PAA were measured using a gel permeation chromatography (GPC) system. It was found that the Mn, Mw, and PDI values of PAA samples with different copolymerization ratios were almost consistent (Appendix A). The thermal imidization process was carried out with the following program: curing at 120 °C for 1 h, 180 °C for 1 h, 240 °C for 2 h, 280 °C for 2 h, and 320 °C for 1 h. Detailed synthetic routes and characterizations are provided in Appendix A.

## 3. Results and Discussion

### 3.1. Synthesis and Characterization of PI Films: Thermal and Mechanical Properties

Through the utilization of the dipole moment variation and the absorption of specific wavelengths of infrared spectroscopy under continuous wavelength light, functional group information can be inferred from polymer molecules. Based on the infrared spectroscopy (FTIR) information shown in Figure 1 (more details are shown in Appendix A), the peak at 725 cm^−1^ represents the angular vibration of C=O, that at 1373 cm^−1^ represents the stretching vibration of C-N in the PI, and those at 1726 cm^−1^ and 1776 cm^−1^ represent the in-phase and out-of-phase stretching vibrations of two C=O groups on the pentamer ring in the polyimide, respectively [4,22]. These absorption bands are considered characteristic absorption peak positions of PI, which indicate the relatively complete progress of the polyimide imidization process. Additionally, the peak value variations in the C-O-C and C-F functional groups specific to the BABP and TFMB monomers, respectively, also coincide well with different copolymerization ratios.

As shown in Appendix A, the XRD data of the PI films exhibit a broad amorphous morphology with scattering peaks at 2θ ≈ 10–20°. The calculated d-spacing values are 5.35, 4.54, 5.21, 4.7, 4.21, 5.02, 4.87, and 5.42 for PI-0%, PI-20%, PI-40%, PI-50%, PI-60%, PI-80%, and PI-100%, respectively, which are determined by the peak value at 18.9 to 16.3 on the corresponding curve according to Braggs equation. The increase in d-spacing means that as the TFMB content increases, the chain arrangement of the copolymerized PI becomes looser, thereby expanding the chain spacing, which is also conducive to the arrangement of polar dipoles under the action of an electric field. As illustrated in Appendix A, as the TFMB content increases, the color of the film gradually becomes lighter, indicating that the charge transfer complex (CTC) effect in the polymer chain is weakened in the presence of a polar unit [31,32]. Meanwhile, the SEM (scanning electron microscope) section image (Appendix A) shows that the PI films have good microscopic morphology, and all samples have a dense and uniform structure without any pores or defects, ensuring that the impact of defects on dielectric properties is eliminated.

We employed Dynamic Mechanical Analysis (DMA) to gain insight into mechanical properties (details included in Appendix A), and the stress–strain curves of PI with different copolymerization ratios were tested (Figure 2b). Among these, the proportion with the lowest tensile strength is PI-0%, with an average tensile strength of only 87 MPa and an average tensile modulus of 0.9 GPa, and the number of flexible groups (C-O-C) is the highest, leading to lower mechanical strength. On the other hand, PI-100% has an average tensile strength of 113 MPa and a tensile modulus of 1.6 GPa. The large-volume side group in this proportion inhibits group flipping and movement, resulting in excellent mechanical properties compared to PI-0%. The copolymer proportion with the best tensile-to-break performance is the PI-60% film, which can reach up to 130 MPa in tensile strength and 2.1 GPa in tensile modulus. Compared to PI-0%, the other five copolymer proportions show varying degrees of improvement in tensile strength and elongation at break. This can be attributed to the balanced changes in the amounts of rigid and flexible groups during copolymerization. Additionally, the introduction of large-volume groups and the reduced molecular chain spacing due to copolymerization contribute to increased density and improved mechanical properties [33].

In order to characterize the thermal and mechanical properties of the samples, we employed DSC (Differential Scanning Calorimetry) (Appendix A) and TGA (Thermogravimetric Analysis) (Figure 2c), and the corresponding statistical data are presented in Appendix A. After compounding TFMB and BABP, the glass transition temperature (Tg) increases (Appendix A). During the polymer heating process, viscoelastic properties change, and a sharp decrease in the storage modulus and loss modulus occurs near the glass transition. The intersection of the tangents of the curves represents the temperature-dependent curves of the storage modulus and loss modulus from DMA tests for the PI-0%, PI-20%, PI-40%, PI-50%, PI-60%, PI-80%, and PI-100% films. The Tg values for these films are 217 °C, 224 °C, 234 °C, 236 °C, 253 °C, 240 °C, and 219 °C, respectively. As the content of the diamine monomer TFMB increases, the Tg of the polyimide first increases and then decreases. The Tg values for PI-0% and PI-100% are 217 °C and 219 °C, respectively. The Tg values of the five copolymer PIs are all higher than those of PI-0% and PI-100%, with the highest Tg of 253 °C observed for PI-60%. This phenomenon is likely due to the introduction of two diamine monomers, which results in more complete cross-linking reactions and higher chain packing density. Additionally, the introduction of rigid structural units (such as C-O-C and -CF3) in the monomers increases the stiffness of the main chain. The chemical cross-linking produced by the molecular chains also severely restricts intermolecular movement, making it difficult for the molecular chains to move at the same elevated temperature, thus increasing the Tg of the copolymer PI [34].

Under a N_2_ atmosphere, the mass loss in PI films was tested from 25 °C to 800 °C. The decomposition temperature at which 5% of the mass is lost is the initial decomposition temperature (Tw, 5%). The percentage of the remaining mass at 800 °C is calculated relative to the original sample mass (WT, 800 °C). The Tw, 5% values for PI-0%, PI-20%, PI-40%, PI-50%, PI-60%, PI-80%, and PI-100% are 447 °C, 448 °C, 436 °C, 403 °C, 451 °C, 485 °C, and 489 °C, respectively. The initial temperature represents the threshold at which PI begins to undergo thermal decomposition, which is above 400 °C for all samples, indicating high thermal stability and structural integrity over a wide temperature range. The WT, 800 °C values are 72.3%, 62.3%, 55.6%, 53.8%, 56.5%, 68.8%, and 50.3%, respectively. Higher WT, 800 °C values indicate slower decomposition rates, which are strongly related to the molecular chain structure. There is a 38 °C difference in Tw, 5% and a 22% difference in WT, 800 °C between the two homopolymers, PI-0% and PI-100%.

In the structure of PI-0%, a large number of aromatic groups form large bonds, raising the decomposition temperature threshold. The C=O bonds in rigid groups have a high bond energy, leading to a high decomposition temperature and a significantly higher WT, 800 °C compared to other copolymers. The introduction of fluorine groups (-CF_3_), due to the high electronegativity of “-F”, results in a high bond energy and resistance to thermal decomposition, giving PI-100% the highest Tw, 5%. The introduction of the two monomers allows the copolymer PI to effectively combine the advantages of different groups, maintaining good thermal stability [35].

In addition to the influence of group structure on the thermal stability of PI, the internal molecular chain structure, combination, and arrangement also significantly affect the thermal properties of PI [36]. This is the main reason for the different thermal stabilities observed in copolymerized PI.

### 3.2. Electrical Insulation

The conjugated structure of the benzene ring in PI and the lone pairs of electrons on O and N in the imide ring contribute to the high electron mobility of PI, leading to electron polarization [37]. Due to the small contribution of electron polarization, the dielectric constant of PI is mainly determined by the dipole polarization contributed by polar groups [5]. The dipole moments of the introduced C=O and C-O-C groups in the BABP monomer are 8.02 μ/10–30 C.m and 2.87 μ/10–30 C.m, respectively [38]. Under an external electric field, these polar groups rotate and orient in the direction of the field, resulting in good dipole polarization, as shown in Figure 3. PI-0% has the best dielectric properties among all proportions, with a ξγ of 4.5 at a frequency of 1 kHz, while that of PI-100% is only 3.4, which is because the nonpolar group -CF_3_ in the TFMB diamine monomer is less prone to polarization under the action of an external electric field and has a slower response to the electric field. It is worth noting that as the content of the TFMB diamine monomer decreases in the copolymer proportion, the dielectric constants of the copolymer PI increase to varying degrees: PI-80%, PI-60%, PI-50%, PI-40%, and PI-20% have values of 4, 4.2, 3.9, 4.3, 4.3, and 4.5, respectively (Figure 3a). The introduction of the BABP monomer forms a conjugated structure between the -C=O and the benzene ring, which is beneficial for electron movement and polarization [39]. This characteristic also allows it to maintain good thermal stability and polarization performance over a wider temperature range, indicating that the copolymer effectively integrates the advantageous functional groups carried by the BABP diamine monomer.

Figure 3b shows the tand of PI films at different proportions. At a frequency of 1 kHz, the tand values for PI-0%, PI-20%, PI-40%, PI-50%, PI-60%, PI-80%, and PI-100% are 0.01, 0.004, 0.008, 0.007, 0.008, 0.006, and 0.006, respectively, all of which are ≤1%. Due to the introduction of the bulky -CF_3_ group, PI-100% has a lower degree of polarization, resulting in lower energy loss from polarization. Additionally, the bulky group acts as a deep trap during charge transfer, capturing dispersed charges, which improves carrier mobility and current density, reduces charge accumulation, and lowers energy loss [40,41]. Therefore, PI-100% has the lowest tand. In contrast, PI-0% has a higher degree of dipole polarization, stronger absorption of the electric field, and higher energy loss, resulting in a higher tand. The simultaneous polycondensation reaction of the two monomers achieves complementary advantages between the monomers, which not only enhances certain properties but also, to some extent, reduces tand. However, at higher frequencies, polarization cannot follow the rapid changes in the external field, leading to a significant increase in loss [8], which is why the dielectric loss at high frequency is higher than that at low frequency.

As shown in Figure 3c,d and Table 1, the E_b_ of PI films is tested at 25 °C and 150 °C, respectively. At 25 °C, the Eb of PI-100% reaches 450 MV/m, which is much higher than that of PI-0% (211 MV/m). When the two diamine monomers are copolymerized with the dianhydride monomer, they exhibit complementary advantages. The E_b_ of the copolymerized films is higher than that of PI-0%. PI-60% exhibits an excellent E_b_ value of 493 MV/m. It exhibits optimal overall performance among the various copolymer ratios evaluated. At 150 °C (Figure 3d), the Eb of PI-60% reaches 473.6 MV/m, which remains the highest E_b_ value among all the PI films tested across the entire range. We also tested the high-frequency (5 GHz) ε_r_ and tanσ of PI films at 25°C (Appendix A), with all the PI films’ ε_r_ values being about 3.0.

The improvement in energy storage performance in copolymer films can be attributed to the increase in E_b_ [42]. According to the free volume breakdown theory, charges are accelerated within the free volume [43]. The E_b_ of the polymer depends on the maximum free volume length, as indicated in Formula (1), where e represents the electron charge, and *E_μ_* represents the breakdown potential barrier of the polymer. lE represents the average length over which the charge is accelerated within the free volume.(1)Eb=EμⅇlE

In PI-0% films, the introduction of C-O-C increases the rigidity of PI but still maintains a relatively large free volume. It is worth noting that copolymerizing two kinds of diamine monomers effectively reduces the spatial free volume and enhances breakdown.

### 3.3. Energy Storage Performance

As shown in Figure 4a, the Ue and charge–discharge efficiency (η) of PI-0% films are very low at 0.79 J/cm^3^ and 86%, respectively. In contrast, the introduction of -CF_3_ significantly reduces the internal free volume of the polymer chain, so that the Ue and η of PI-100% are approximately 3.7 J/cm^3^ and 80%, respectively. According to the formula for calculating the Ue of linear polymers (Appendix A), Ue is directly proportional to the square of ξr and Eb [44]. It is evident that improving Eb plays a crucial role in increasing the Ue of the polymer, which aligns very well with our data results. The Ue and η of PI-60% are 5.07 J/cm^3^ and 82%, respectively. Moreover, copolymerization with several other components yields an energy density of about 2 J/cm^3^, higher than that of PI-0%. Random copolymerization can improve the internal defects of polymer molecules by improving the rigidity of the molecular structure, and can finally form a compact molecular structure to improve the insulation properties of polymers [45]. Notably, as shown in Figure 4b, an increase in the test temperature to 150 °C leads to an unexpected enhancement in the Ue of PI-0%, which rises to 1.42 J/cm^3^ compared to its value at 25 °C. This improvement is hypothesized to result from the incorporation of the C=O group as a rigid structural component, thereby augmenting the thermal insulation properties of PI under elevated temperatures, while PI-60% can still achieve a discharge energy density of 2.5 J/cm^3^, which is higher than that of other films. As shown in Appendix A, the tans of PI films at 25 °C and 150 °C under 100 MV/m are still low. As shown in Figure 4c, the Ue values of PI films with different copolymer ratios at 25 °C exceed those of other modified PIs [21,24,46,47,48,49,50] and are even higher than those of composite polymers [51,52,53], showing that our random copolymerization scheme represents great progress in the modification of intrinsic PI.

We plotted these properties on a radar chart (Appendix A) to more intuitively visualize the performance trends across different ratios. The larger the area enclosed by the lines connecting the performance metrics, the better the overall performance. Based on this, we concluded that PI-60% exhibits optimal performance.

In this paper, the DMol3 computational module was used to calculate energy levels and neighboring functions. Based on DFT, the electrostatic potential distributions of three monomers, TFMB, BTDA and BABP, were obtained (Figure 5a–c). Compared to BABP, which has a more uniform potential distribution, TFMB effectively separates positive and negative charges on its main and side chains. This separation of charges results in deep traps that capture and disperse charges, thereby enhancing charge carrier mobility, current density, and Eb. Additionally, the local reactivity of atoms within the molecules, as shown by Fukui functions (Appendix A), indicates that the C=O group in BABP exhibits higher reactivity than other positions. This is beneficial for improving the dipole orientation and εr of the cross-linked system [54].

The reactivity of these monomeric molecules is primarily influenced by the electronic effects of different substituents [33]. The nucleophilicity of a monomer is related to its Highest Occupied Molecular Orbital (HOMO), while its electrophilicity is associated with its Lowest Unoccupied Molecular Orbital (LUMO) [55]. As shown in Figure 5d and Appendix A, it can be inferred that these differences are also reflected in the random copolymerization, leading to different cross-linking tendencies at different copolymer ratios. In the calculation results in Figure 5e, it is observed that the van der Waals forces (electrostatic) exhibit a regular decrease (increase) with varying copolymer ratios. However, this phenomenon was not subject to further investigation in the present study. The cohesive energy (Ecoh) mentioned in this paper mainly takes into consideration the sum of van der Waals forces and electrostatics. With varying ratios, significant differences in Ecoh are observed which directly affect the thermal and mechanical properties of the PI films [23]. Specifically, when the polymer ratio is PI-60%, Ecoh is the highest (1.26 × 10^9^ J/m^3^), indicating the best thermodynamic performance.

In the data of UV-vis-NIR summarized in Figure 5f, all PIs exhibit absorption peaks around λg < 300 nm, which are attributed to the electronic transitions resulting from the absorption of double bonds in the aromatic ring structure. In the wavelength range around 350 nm, the absorption peaks of PI-20%, PI-50%, and PI-80% show a significant redshift phenomenon; it is speculated that this redshift is due to the absorption bands of characteristic functional groups such as C=O and -N-O, which form large π bonds in the copolymer system, and this reduces the energy difference between energy levels and consequently decreases the energy of electronic transitions [22,56]. This result is consistent with the decrease in breakdown strength observed in the experimental tests.

However, in the PI-60% films, Eg did not decrease. This may be attributed to an optimal ratio of polar functional groups to π-conjugated molecules leading to strong polarization at this copolymer ratio, which induces strong polarization. Consequently, electron transitions become more challenging, stabilizing both the cationic and anionic states, thereby achieving a more energetically stable state [20,57].

## 4. Conclusions

In conclusion, this study presents an innovative molecular design strategy for enhancing the intrinsic properties of polyimide materials toward advanced energy storage capacitor applications. The developed random copolymerization methodology enables the effective integration of complementary characteristics from two distinct diamine monomers, resulting in the synergistic enhancement of dielectric properties. Under the optimal ratio (PI-60%), the copolymer exhibits high T_g_ (263 °C), high tensile strength (120 MPa), high ε_r_ (4.2), high E_b_ (493 MV/m), and low tanδ (≤0.008). Ultimately, U_e_ and η are 5.07 J/cm^3^ and 82%, respectively. Furthermore, based on DFT calculations, the copolymerization tendencies of the PI films were verified by electrostatic distribution simulations, calculations related to energy levels and nearby functions, and UV-vis-NIR. It was confirmed that the PI with a PI-60% copolymerization ratio is the most stable and exhibits the best overall performance, aligning perfectly with the experimental results. These experimental findings provide a basis for unlocking the infinite possibilities of intrinsic PI in capacitors.

## Figures and Tables

**Figure 1 polymers-17-00753-f001:**
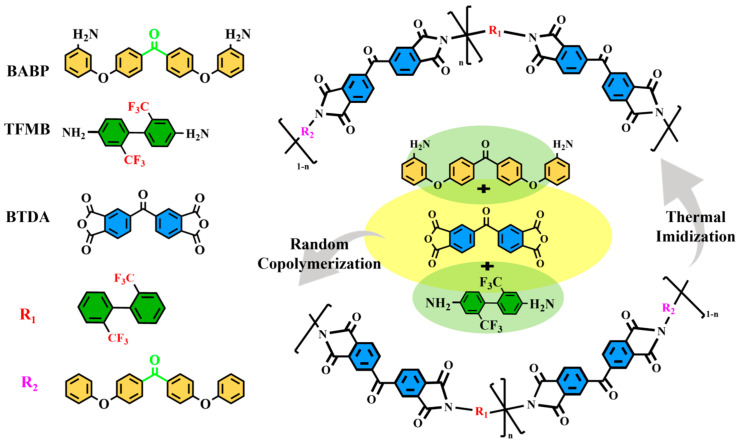
Schematic of the synthesis process for PI.

**Figure 2 polymers-17-00753-f002:**
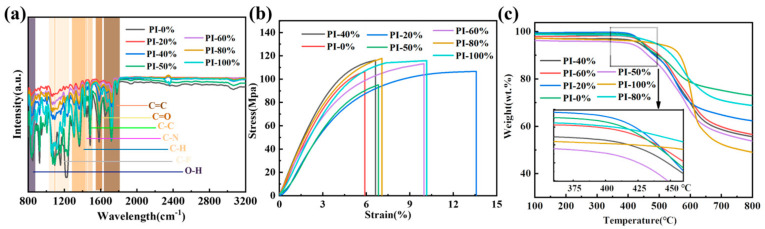
(**a**) FTIR, (**b**) strain–stress curve, and (**c**) TGA of PI films in different copolymerization systems.

**Figure 3 polymers-17-00753-f003:**
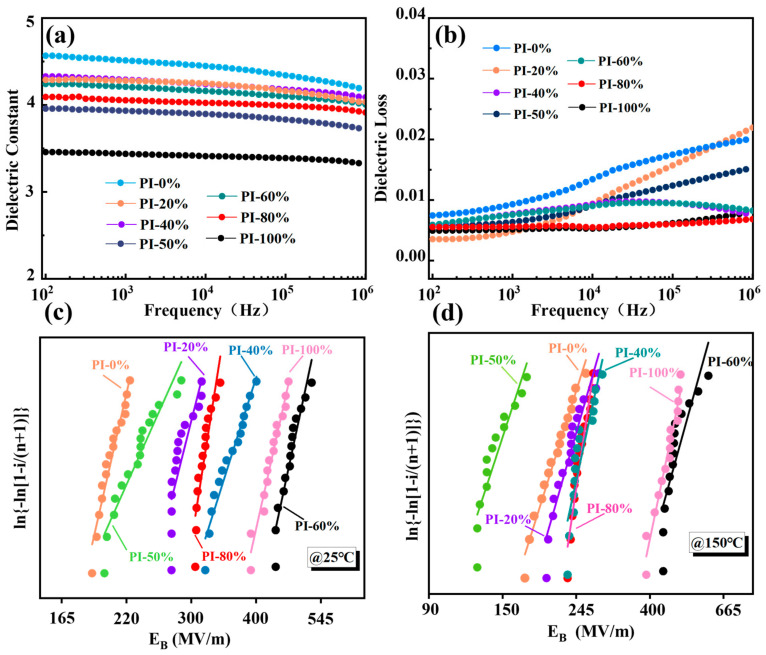
(**a**) The dielectric constant, (**b**) dielectric loss, and Weibull breakdown strength of polyimide films with different copolymer ratios at (**c**) 25 °C and (**d**) 150 °C.

**Figure 4 polymers-17-00753-f004:**
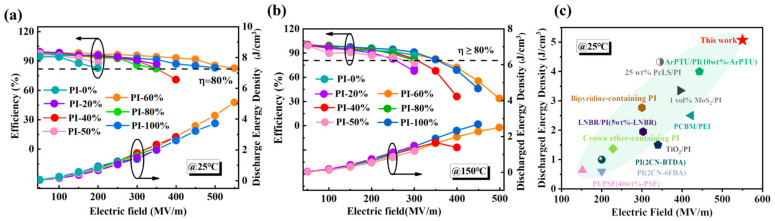
The discharge energy density and charge–discharge efficiency of PI films with different copolymer ratios at (**a**) 25 °C and (**b**) 150 °C. (**c**) Comparison of discharged energy density for pure polymers at 25 °C.

**Figure 5 polymers-17-00753-f005:**
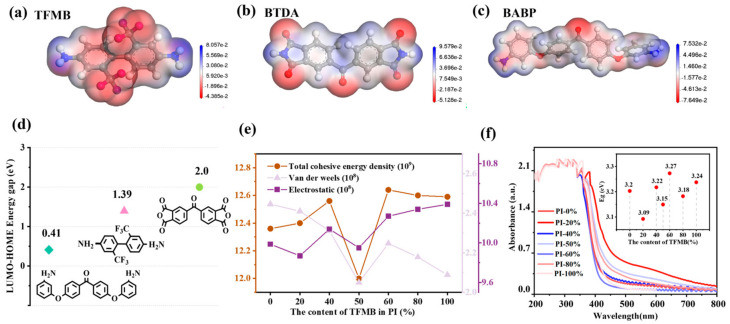
(**a**–**c**) The electrostatic potential distribution of three monomers according to DFT calculations. (**d**) The energy gap difference between HOMO and LUMO for three monomers was calculated. (**e**) The cohesive energy, van der Waals forces, and electrostatics were studied by DFT calculations. (**f**) UV-Vis-NIR diffuse reflectance map of PI films with different copolymerization ratios: the small graph in the upper right corner shows the absorption wavelength of different polymer ratios and the optical bandgap (E_g_) calculated using Planck’s Law.

**Table 1 polymers-17-00753-t001:** The E_b_ and β of PIs with different copolymerization ratios at 25 °C and 150 °C.

	PI-0	PI-20%	PI-40%	PI-50%	PI-60%	PI-80%	PI-100%
E_b_ (25 °C)	247	295	372	211	493	323	450
b (25 °C)	9	16	13	19	20	19	18
E_b_ (150 °C)	186	217	264	144	474	260	434
b (150 °C)	9	10	13	12	10	19	13

## Data Availability

The original contributions presented in this study are included in the article/Appendix A. Further inquiries can be directed to the corresponding author.

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
