# Peer review of "Polyimide Modified with Different Types and Contents of Polar/Nonpolar Groups: Synthesis, Structure, and Dielectric Properties"

_polymers, 2025, doi:10.3390/polym17060753_

Round 1
Reviewer 1 Report
Comments and Suggestions for Authors
In this work, the authors reported an indiscriminate copolymerization method was proposed, which utilizes 13 two different diamine monomers with bulky side groups (-CF3) and high polarity (C-O-C) respectively, to copolymerize with the same dianhydride monomer and prepare a series of intrinsic PI films, and the PI films with highly dipolar rigid backbone maintains excellent thermal and mechanical properties while enhancing the dipole polarization. Under the optimal copolymer ratio, the dielectric constant and dielectric loss are 4.2 and 8‰, respectively. The author reported at room temperature, the highest breakdown strength reaches 493MV/m, and the energy storage density and charge-discharge efficiency are 5.07 J/cm3 and 82%, respectively. Many experiments/tests were conducted, but some results and observations were not sufficiently discussed and presented. The manuscript needs major revision before being considered for publication. Some specific comments are listed below
1. How were the the seven different copolymerization ratios of PAA chosen (Table S1)? The author give some hint or identify trends.
2. Page 3 line 114”….’shown in Table.1…to…..Table S1” should be corrected to indicate that these are placed in SI.
3. Page 3 line 154 “…. the TFMB content increases, the color of the film gradually becomes lighter, indicating that the charge transfer complex (CTC) effect in the polymer chain is weakened in the 155 presence of polar units….” This scientific justification is insufficient and should be supported by references to relevant scientific papers to strengthen the argument and provide evidence for the observed phenomenon.
4. Page 3 line 157 “ ……all samples have a dense and uniform structure without any pores or other defects….” This claim at 10-micrometer bar length magnification can provide a general overview of the surface but is not sufficient to conclusively determine the presence or absence of pores. Therefore, the author should conduct high resolution imaging to support their claim.
5. The authors conducted the stress-strain of PIs with different copolymerization ratios and only present PI -60 and PI-0. All result should be discussed and explained in detail. It seems like report than a discussion. Most importantly, the author didn’t mention which ASTM standard was used to measure the samples?
6. Page 3 line 171 “….the elevation of Tg ofcPI-60% and PI-40% can be attributed to the increased chain packing density, which hinders molecular chain mobility…” A detail justification is needed. Furthermore, an explanation of what happened at 263°C is required to provide the readers a clear understanding at every transition.
7. In Figure 3b, the dielectric loss of PI-60 presents a different trend compared to other copolymerization ratios. Why?
8. Page 5 line 216 “…….It is worth noting that copolymerizing two kinds of diamine monomers effectively reduces the spatial free volume and enhances breakdown.” This final remark should be supported by relevant scientific papers.
9. The author reported the discharge energy density and charge-discharge efficiency of PI films with different copolymer ratios at (a) 25 °C and (b) 150 °C (Figure 4) with a single data. Thus, the author should replicate at least five times and report the results with a standard deviation error bar for all tabulated data presented in the paper.
10. Finally, I recommend that the authors includes more recent scientific findings to support the results and discussion.

Reviewer 2 Report
Comments and Suggestions for Authors
The authors have presented a systematic study that investigates structural and electronic properties of PI-based materials. The manuscript is comprehensive, however lacks specific details and is not consistent throughout in explaining the results and may be accepted following major revisions.
(1) The XRD results provided in the SI and the main text with respect to the d-spacing is inconsistent. Can the authors explain the trend clearly and state how the results can be correlated to the inter-chain spacing/packing and free volume?
(2) Figure 2b and lines 160-165 page 4: There is not a very clear trend in the modulus and Pi-60% doesn't have the highest value necessarily both in the linear regime (Young's modulus) and the tensile modulus. Can the authors explain the results clearly and also complete the explanation in the text "It is probable that the BABP and TFMB monomers."
(3) How can the free volume be quantified in this work given that it has been used often to correlate with the electronic properties and stability?
(4) Can the authors comment briefly on the rationale for why their polymers perform better in comparison to unmodified analogues or state-of-the-art systems and how the functional group modifications/additions offer an advantage structurally and for electronically?
(5) W.r.t DFT calculations, are the energies (van Der Waals, electrostatic plotted as absolute values)? Also the authors mention about using their properties for informing machine learning models, however they haven't concluded which properties are most important from their analysis and study. Can they comment on that and how feasible it is to use these energy calculations for a large subset of systems?
(6) The authors must provide appropriate references for DFT methods? Also they must mention why they chose the specific functionals and what are the error bars associated with their calculations?
Reviewer 3 Report
Comments and Suggestions for Authors
The manuscript "Polymide Modified with Different Types and Contents of Polar/No-polar Groups: Synthesis, Structure and Dielectric Properties" shows combination of PI with two different diamine monomers to enhance polarized groups in backbones with achieve energy density. The content in view of science is well written. There are several sentences that need be corrected in manuscript as well supplementary.
1. What are the main application for more dense PI composite. Please add those in the introduction as well conclusions.
2. The PI 60% authors says have best results of other PI composite. What are the reason for such? From table S3 the PI-80% has decrease in young's modulus but increase in tensile strength. Please explain this more clear.
3. There are some abbreviations not defined where they first appear such as: PEI and PCMB in page 2 line 55. Also there several typos as well sentences that are too long and difficult to understand or even some parts are missing.
Page 2, line 94: "In this article..." This sentence is too long and difficult to understand. Please made 2 out of it.
Page 3 line 100: "8%o" please correct that.
Page 4 line 160 "we" must be We
Page 4 line 164. "It is probable that the BABP and TFMB monomers." the sentence incomplete, please correct that
Page 5 line 188 - 191. The two sentences have grammar issues also those are difficult to understand. Please reformulate
Page 7 line 270. "Electrostatic" what electrostatic and what values of such. Please give a more clear expression.
There are several more of those sentences, please check English and grammar and if some of those meanings intended are obtained.
Comments on the Quality of English LanguageThe English is not good and should be corrected in grammar and spells must be checked. A native speaker should check the manuscript as there several sentences where contents are missing.
Round 2
Reviewer 1 Report
Comments and Suggestions for Authors
The authors did a good correction and respond to the questions adequately. However, a few concerns regarding question 5, and I recommend it to be published in polymer after addressing the following minor revision.
The authors still have not present ASTM or any other-well known standard procedure for the stress-strain analysis.
Reviewer 2 Report
Comments and Suggestions for Authors
The authors have addressed all the comments thoroughly and provided insights to significantly improve the manuscript quality. The manuscript can be accepted for publication in its current form.
